# Recent Advances in Genome Editing and Bioinformatics: Addressing Challenges in Genome Editing Implementation and Genome Sequencing

**DOI:** 10.3390/ijms26073442

**Published:** 2025-04-07

**Authors:** Hidemasa Bono

**Affiliations:** 1Graduate School of Integrated Sciences for Life, Hiroshima University, 3-10-23 Kagamiyama, Higashi-Hiroshima 739-0046, Japan; bonohu@hiroshima-u.ac.jp; Tel.: +81-82-424-4013; 2Department of Biological Science, School of Science, Hiroshima University, 3-10-23 Kagamiyama, Higashi-Hiroshima 739-0046, Japan; 3Genome Editing Innovation Center, Hiroshima University, 3-10-23 Kagamiyama, Higashi-Hiroshima 739-0046, Japan

**Keywords:** genome editing, next-generation sequencers, genome sequencing, bioinformatics, bibliome, transcriptome, meta-analysis, pathway, database

## Abstract

Genome-editing technology has advanced significantly since the 2020 Nobel Prize in Chemistry was awarded for the development of clustered regularly interspaced short palindromic repeats (CRISPR) and CRISPR-associated protein 9 (Cas9). While CRISPR–Cas9 has become widely used in academic research, its social implementation has lagged due to unresolved patent disputes and slower progress in gene function analysis. To address this, new approaches bypassing direct gene function analysis are needed, with bioinformatics and next-generation sequencing (NGS) playing crucial roles. NGS is essential for sequencing the genome of target species, but challenges such as data quality, genome heterogeneity, ploidy, and small individual sizes persist. Despite these issues, advancements in sequencing technologies, like PacBio high-fidelity (HiFi) long reads and high-throughput chromosome conformation capture (Hi-C), have improved genome sequencing. Bioinformatics contributes to genome editing through off-target prediction and target gene selection, both of which require accurate genome sequence information. In this review, I will give updates on the development of genome editing and bioinformatics technologies with a focus on the rapid progress in genome sequencing.

## 1. Introduction

Where are we now with genome-editing technology? It has been four years since the Nobel Prize in Chemistry 2020 was awarded for the development of a method for genome editing. Since then, genome-editing technologies have become more widely used. During that time, the organelle genome-editing technology represented by clustered regularly interspaced short palindromic repeats (CRISPR) and CRISPR-associated protein 9 (Cas9) has been widely used in the academic research field [1,2,3,4,5,6,7,8,9,10,11].

However, social implementation of genome-editing technology has not progressed [12]. This is partly because the CRISPR–Cas9 patent dispute has not been settled, and may also be that the analysis of gene function has not progressed as much as the development of genome-editing technology. Therefore, the development of approaches that bypass direct gene function analysis is eagerly awaited. Bioinformatics and next-generation sequencing (NGS) will be key [13].

First, how we use NGS is crucial. It is essential to sequence the genome of the species whose genome you want to edit. Open access data on reported genome sequencing are organized in the NCBI Datasets database (https://www.ncbi.nlm.nih.gov/datasets/ (accessed on 1 April 2025)). However, some data are not at a reusable quality level, or some archived data are flagged as contaminated even though they are archived at NCBI. Therefore, as of 2025, there is a need to decode the genome of the target species itself, and it is not always possible to sequence the genome of any species. Many problems exist, such as the heterogeneity of genome sequences, ploidy, and the small size of individuals. In the late 2010s, it was said that short- and long-read hybrids were the basic method for sequencing new genomes, but now, in 2025, it is mostly possible with PacBio high-fidelity (HiFi) long reads [14] and high-throughput chromosome conformation capture (Hi-C) [15].

Another important technology is bioinformatics. There are two contributions that can be made to genome editing:Off-target prediction;Selection of target genes.

However, these cannot be carried out without genome sequence information for the target organism.

In a previous review in 2022 entitled “Genome editing and Bioinformatics” [16], we summarized the development of genome-editing technologies and bioinformatics to date. This current review updates the previous one and outlines not only bioinformatics technologies for genome editing, but also those for genome sequencing, with a particular focus on the rapid progress of genome sequencing (Figure 1).

## 2. Genome Sequencing

There are two main types of genome analysis. One is sequencing DNA fragments and mapping these to a reference genome that has already been determined. The other is de novo assembly by decoding the genome sequence. De novo genome analysis makes it possible to study species whose genomes have never been decoded before and to search for their genes. NGS is indispensable for the sequencing of the genome.

Since the completion of human genome sequencing, various NGS methods have been developed, but the most widely used NGS methods are shown in Figure 2. There are two types of NGS methods: long reads and short reads. Short reads are sequencers that can decode tens or hundreds of millions of reads of contiguous DNA sequences of a few hundred (100–300 bases) in length. Long reads, on the other hand, are capable of sequencing DNA sequences of tens of thousands or more bases in length. Short reads can read many reads with high sequence quality but short sequencing lengths, while long reads can read long sequences but their sequence quality is considered to be low.

### 2.1. Application of Next-Generation Sequencers

The PacBio HiFi read technology has been developed to provide long reads with high sequence quality [14]. Table 1 shows genome sequencing targets and their statistical values in the Hiroshima University Genome Editing Innovation Center, and several sequencers have been employed to sequence genomes. Currently, the PacBio HiFi read in conjunction with Hi-C, which provides information on chromosomal proximity, are de facto standards.

#### 2.1.1. HiFi Reads by PacBio Sequencing

Long-read sequencing developed by PacBio is referred to as SMRT (single molecule, real-time) sequencing, and is performed on single molecules. Simply put, it is carried out by observing the DNA polymerase elongation reaction of a single molecule of DNA dropped into a microscopic well (known as a zero-mode waveguide; ZMW) [17]. The elongation reaction (sequence reaction) can be performed until the DNA polymerase is inactivated (24 to 30 h), and as a result, it is possible to sequence long reads that are longer than short reads. A movie of this sequencing technology is available on YouTube (https://youtu.be/NHCJ8PtYCFc (accessed on 1 April 2025)).

PacBio Sequel IIe (PacBio, Menlo Park, CA, USA) uses a cell with about 8 million wells (ZMWs), while PacBio Revio (PacBio, Menlo Park, CA, USA) uses four cells with 25 million ZMWs in each cell, and sequence reactions are observed in parallel. The Revio has about three times as many ZMWs and four times as many cells as the Sequel IIe, which means that twelve times as many sequence reactions can theoretically be observed.

The reads obtained are called continuous long reads (CLRs), and the accuracy at this point is about Q10 (85–90%). The DNA is cycled by attaching adapters to both ends of the double-stranded DNA, and the DNA is sequenced repeatedly. Error correction is performed using the reads sequenced multiple times to obtain circular consensus sequence (CCS) reads, and this correction yields an accuracy of about Q20 (about 99%). The key is to decipher the sequence repeats within the same molecule. In SequelII/IIe and Revio, consensus sequences are obtained from the decoded sequences, resulting in single-molecule-long reads with an average read accuracy of Q33, known as HiFi (High-Fidelity) reads. As a result, HiFi reads yield read lengths of 15,000 to 20,000 (15–20 k) bases with a sequence quality higher than the 1 in 1000 base error precision (Q30).

In actual sequencing, it is very important to take long DNA (15 k–20 kbp) and to sequence at least 30X the estimated genome size for genome sequencing.

#### 2.1.2. Hi-C

Unfortunately, neither the short-read nor the long-read technologies described above will be able to sequence an organism’s chromosomes from end to end in 2025. In the case of the longest human chromosome, a very long sequence of about 250 million (250 M) bp is cut into pieces to a convenient length for sequencing. The sequence is then connected into a single strand on a computer (genome assembly, which will be explained in detail in the next section) to obtain a genome sequence that is virtually a single strand for each chromosome.

The DNA sequence that makes up the genome is a single strand of DNA folded into each chromosome, which together with proteins make up a structure called chromatin in eukaryotes. To use an analogy, chromatin is like a silkworm moth cocoon. The silkworm moth cocoon also consists of a single silk thread that surrounds the pupa like an eggshell. It is easy to imagine that the silk threads may be far away from the pupa, but in the three-dimensional structure, they are very close to the pupa. Chromosomes are structured in the same way as cocoons, and information on DNA sequences near each other can be measured using a different experimental method. This is Hi-C (short for high-throughput chromosome conformation capture), a high-throughput version of chromosome conformation capture [3]. Hi-C is a method of finding regions in the chromatin structure by attaching (ligating) adjacent parts of the structure through chemical processing and then finding those regions by DNA sequencing. Hi-C was originally developed for the study of chromosome structure. The method has come to be used because the three-dimensionally close regions on the chromosome obtained by Hi-C are useful for genome assembly as sequences on the same chromosome.

While not essential in genome assembly, Hi-C is often used in large genome assemblies where the genome size exceeds several gigabases (Gb) in length in 2025 genome sequencing. Dovetail^®^’s AssemblyLink, which uses endonucleases for chromatin fragmentation in proximity ligation assays, is often used because it provides uniform coverage across the entire genome without depending on restriction enzyme sites (https://cantatabio.com/dovetail-genomics/products/dovetail-assemblylink-kit/ (accessed on 1 April 2025)). It is estimated that a total of about 100 Gb of short reads are required, and the sample must contain not only the DNA itself, but also tissue that retains chromatin structure.

It is standard practice to take samples from a single individual for genome sequencing. This is because genome assembly often fails when multiple individuals are used due to their genetic diversity. Sampling does not always work. This is a major misconception about genome sequencing: there is no one way that will always work. The extraction method must be changed according to each situation, and this is a rule of thumb.

### 2.2. Transcriptome Analysis

There are often cases where the genome cannot be sequenced because the genome size is too large and/or the amount of genome from one individual is too small. In these cases, the transcriptome should be analyzed. The decoded nucleotide sequences that are thought to be derived from the same gene are combined into a single sequence for which bioinformatics is indispensable. This is presented further in the next section.

## 3. Bioinformatics

Bioinformatics is a field that has been used since the late 1990s as a fusion of biology and information science [18]. Bioinformatics is an academic field of study that aims to develop methodologies and software using information science, statistics, and other algorithms for the various pieces of “information” possessed by living organisms, and to elucidate biological phenomena (in silico analysis) through analysis using these methods. Bioinformatics is a term that refers to such a wide range of fields and has various definitions. In a broad sense, “bioinformatics” refers to all life science data analysis using computers, and in a narrow sense, genome sequence analysis (Figure 3).

In other words, the analysis which is carried out after decoding the nucleotide sequence data is bioinformatics (in the narrow sense). In the series of genome analysis processes outlined in Figure 1, the gray shaded areas are the parts that cannot be implemented without bioinformatics, i.e., genome analysis is impossible without bioinformatics. The following is a step-by-step description of where bioinformatics plays an active role in genome analysis.

### 3.1. Public Database Surveys

First, it is necessary to check whether the genome of the target organism has already been sequenced and registered in a public database. Even if the genome has not been sequenced, it is possible to estimate the genome size of the target organism from that of a closely related species by checking whether the genome of the closely related species has been sequenced. For example, the genome size of humans (*Homo sapiens*) is around 3.1 Gb and that of chimpanzees (*Pan troglodytes*) is around 3.2 Gb, very close in size. However, there is no guarantee that the genome sizes of closely related species are the same. For example, the genome size of the brown planthopper (*Nilaparvata lugens*) is around 1.1 Gb while that of closely related planthoppers (*Sogatella furcifera* and *Laodelphax striatellus*) is about half (around 0.5 Gb).

NCBI Datasets, one of the NCBI databases, is often used to check whether a genome sequence has been decoded and registered (https://www.ncbi.nlm.nih.gov/datasets/ (accessed on 1 April 2025)). The NCBI Datasets database contains detailed data for species with the same scientific name that have been sequenced and assembled by different groups (referred to as assemblies). Data where contamination has been detected are marked (flagged). The reason why such data are not removed but remain marked is probably because NCBI is a part of the National Library of Medicine (NLM), which archives data. Although such marked data can be downloaded, one should be careful about its use, as it may contain sequences of other organisms.

In addition, it is necessary to look carefully at the attribute information of the registered organisms. This is because in many cases the species is the same, but the strain is different, and even if the species is the same, the quality of the genome sequence may not be at the chromosome level. It is important to carefully examine the information in public databases.

### 3.2. Assembly

The largest human chromosome, the human chromosome 1, contains approximately 250 million bases (https://www.ncbi.nlm.nih.gov/datasets/genome/GCF_000001405.40/ (accessed on 1 April 2025)). However, no technology has yet been established to decode the genome sequence of a single chromosome from end to end at a time. Therefore, the genome sequence of each chromosome is constructed by joining many fragment sequences on a computer. This process is known as genome assembly.

To decode a genome sequence by genome assembly, many “glue sites” are required. As a result, the same part of the genome is sequenced many times. Of course, the number of “reads” varies depending on the location, but the average value is called “coverage,” which is defined as following Equation (1).coverage = total number of bases read/(estimated) genome size(1)

For example, in the case of the beefsteak plant (red perilla) in Table 1, the number of bases sequenced was 72.4 Gb and the estimated genome size was 12.6 Gb, which means that the coverage was 72.4/12.6 = 57.5X (Table 1).

A paper comparing 11 different genome assembly programs (called genome assemblers) shows that PacBio HiFi, the standard in genome sequencing, can obtain a 20–30X read coverage and produce a good genome [19]. In the case of PacBio HiFi, which is the standard for genome sequencing, the benchmark results show that a good genome assembly can be obtained with a 20–30X read coverage. We will first discuss the metrics required to sequence the results of those genome assemblies.

Assembling a genome should ideally lead to a single sequence for each chromosome, but this is not the case. The resulting sequence group, which is the result of connecting as many sequences as possible by the genome assembler, is called a contig. It is possible to make longer and fewer contiguous sequences (called scaffolds) by further connecting them with other pieces of information. There are two indicators of genome quality, namely N50 and BUSCO.

N50 is a common indicator for assessing the quality of a genome assembly and indicates the “contiguity” of the assembly. Specifically, N50 is the length of a sequence (in base pairs) when it reaches half of the total length when the sequences are arranged in order of length and added from the top. Since N50 represents the length of the middle of the total length while looking at the distribution of the obtained sequences, N50 will be large if there are many long sequences. Conversely, if there are few long sequences and many short sequences, N50 will be small. Note that N50 can be calculated for both scaffolds and contigs.

Another indicator is BUSCO (benchmarking universal single-copy orthologs), which is a tool used to check the accuracy of a genome sequence (https://busco.ezlab.org (accessed on 1 April 2025)) [20]. This is accomplished by checking how many core gene sets are present in the assembled sequence. The closer to 100%, the better. It is necessary to select an appropriate DB to be used as a reference depending on the target species.

Various genome assemblers have been developed, each with its own advantages and disadvantages. As of 2025, Hifiasm is the most commonly used genome assembler for species with genomes larger than several hundred megabases [21]. While one study has reported that Flye is better for relatively small genomes (tens to hundreds of megabases) [22], it is important to choose the genome assembler that leads to a better connection among them. The current situation is that the usage of the system is still in its infancy. This is because genome assembly is still a developing field, and no definitive program has yet been established.

An important feature of genome assemblers used in 2025 is the ability to assemble diploid genomes (diploid-aware assemblers). In the past, it was necessary to use cultivars or other methods that made the two sets of genomes as homologous as possible to sequence a genome. However, now even heterozygous genomes can be decoded. A major contribution to this is the proximity information on chromosomes (Hi-C) described in the previous section. Currently, Hifiasm is the most used program for this purpose. With the recent version upgrade, some ONT sequences can also be assembled with Hifiasm, which has been well received.

As mentioned in the previous section, when the genome cannot be decoded, transcriptome sequencing (sequencing RNAs, which is often called RNA sequencing or RNA-seq) is an alternative method, with which sequence data can be assembled. In this case, Trinity is the de facto standard assembly program for transcriptome assembly [23].

### 3.3. Annotation

Annotation refers to the process of putting up a sticky note and adding a description so that anyone can see it later, since it is difficult to tell what is in that location. Unannotated genome sequence data are not suitable for actual use if the genome sequence has been just decoded and connected by genome assembly. Therefore, annotation is very important. There are two major types of annotation: genome annotation and functional annotation. Genome annotation is the direct annotation of gene-coding regions and transcriptional regulatory regions to genome sequence data. On the other hand, functional annotation is the annotation of functions of genes in the predicted gene-coding regions, which started with the functional annotation of mouse (FANTOM) project in RIKEN [24,25,26,27,28,29]. The following sections will introduce the details of each of these two types of annotation.

#### 3.3.1. Genome Annotation

Genome annotation is the mapping of DNA fragments obtained by experimental methods to a reference genome sequence to describe its genomic coordinates and what information they impart. In other words, it annotates the location of genes and other items in the genome. It can be roughly classified into two types: de novo-based and reference-based. Currently, the reference-based method by RNA-seq is more reliable if the RNA-seq reads are deep enough with tens of millions of reads. Its disadvantage is that genes cannot be detected if the gene is not transcribed as mRNA. The de novo prediction is also essential in some cases where reads of RNA-seq are not enough.

The most typical genome annotation is the annotation of gene-coding regions in the genome. Even with complete genome sequences and findings in humans and mice, which have been studied the most as models, computer programs alone cannot predict all gene regions from new genome sequences (as of 2025). Therefore, genome annotation using complementary DNA (cDNA) reverse transcribed from mRNA is still the best solution. As of 2025, the best practice has been to add annotation by de novo gene prediction by mapping cDNA reads to the genome. For eukaryotes, BRAKER (https://github.com/Gaius-Augustus/BRAKER (accessed on 1 April 2025)), a pipeline that automates gene structure prediction of protein-coding genes from novel genome sequences, is often used as a tool for this purpose [30]. BRAKER3 is version 3 of BRAKER, which uses the gene prediction programs GeneMark and AUGUSTUS, as well as RNA-Seq data to increase its accuracy. RNA-seq data can be obtained not only from conventional short-read sequencers, but also from ISO-Seq data obtained from PacBio long-read sequencers.

If the above-mentioned gene-coding regions are well annotated, the transcription start site (TSS) should naturally be determined, but in reality, multiple TSSs are known to exist for a single gene. An experimental method that can be used to measure transcription start sites has been developed, and cap analysis of gene expression (CAGE), also known as CAGE-seq, is a well-known method for this purpose [31]. CAGE is a technique that utilizes the cap structure at the 5′ end of mRNA to cut out approximately 20 bases from the 5′ end as a tag, and in combination with NGS is an extremely sensitive and accurate quantitative transcriptome analysis method. This feature enables the rapid identification of transcription start sites and inference of promoters and transcription factor-binding motifs. It is also possible to measure the expression level of each gene by the number of tags.

In the previous section, we explained that Hi-C is used in genome assembly, but in reality, the original Hi-C is used for finding topologically associated domains (TADs) in the genome [15]. By mapping to the reference genome, the Hi-C map can be obtained as to where the chromosomes were in three-dimensional proximity.

There is also a method for sequencing DNA sequences, using NGS, in the regions where histones and transcription factors bind to the genome, called ChIP-seq (chromatin immunoprecipitation sequencing). Another method, called ATAC-seq (assay for transposase-accessible chromatin using sequencing), selectively sequences the nucleosome-free regions called open chromatin regions to determine the accessibility and proximity of chromatin in the genome sequence. These methods are often used in epigenomic analyses, as well as Hi-C, to study chromosome structure.

A genome browser is used to visualize genome annotation. The UCSC Genome Browser (https://genome.ucsc.edu/ (accessed on 1 April 2025)) and Ensembl Genome Browser (https://www.ensembl.org/ (accessed on 1 April 2025)) are genome browsers available on the Internet. If the genome assemblies are identical, they can be displayed on these genome browsers as a custom track. Even if they are not, several local genome browsers have been developed that allow you to visualize them on your own computer.

-IGV (Integrative Genomics Viewer; https://igv.org/ (accessed on 1 April 2025))-JBrowse (https://jbrowse.org/jb2/ (accessed on 1 April 2025))

The annotation of transposable elements has recently become a hot topic, now that genome sequences of closely related species can be decoded [32].

#### 3.3.2. Functional Annotation

Once the gene-coding regions of a new organism have been predicted, researchers will first want to know the function of those genes. Gene 1 of another organism, which is thought to be derived from the same gene as gene 2, is called a homolog, and gene 1 is said to be sequence homologous to gene 2. Homologs often have similar gene sequences (high sequence similarity) and consequently have similar functions. Therefore, the following inferences can be made.gene 1 has sequence homology with gene 2 + gene 2 has function F -> gene 1 has function F(2)

Attempts have been made to use this logic to predict gene function, but in some cases, even homologs have different functions. For example, epsilon crystallin (PDB entry: 1o9j) in the cape gerbil, a type of crystallin protein that makes up the lens of the eye, also functions as an aldehyde dehydrogenase (https://pdb101.rcsb.org/motm/127 (accessed on 1 April 2025)). Among the homologs, attention has been paid to orthologs, which are defined as genes with homologous functions that have become genes of different organisms through speciation during the course of evolution. In other words, since orthologs are likely to have the same function, ortholog assignment is a method of assigning an orthologous relationship between a gene of a newly sequenced organism species and a reference organism species and predicting its function using this analogy.

Until now, this ortholog assignment has been performed as follows. If, for example, gene A was found to be the best hit in a sequence similarity search for all genes of species 1 using gene P of species 2 as the question sequence, then gene A and gene P were considered to be orthologs. Recently, however, because of the availability of chromosome-level genome sequences and gene sets of closely related species, a method that considers not only the best hit but also the order in which genes are encoded on the chromosome to estimate orthologs, or synteny information, has come into use. This method is now being used.

Gene ontology (GO) is a controlled vocabulary originally created to describe genes in eukaryotes (initially mouse, Drosophila, and budding yeast) [33]. It has three ontologies (biological process, molecular function, and cellular component), and gene functions are annotated in terms of these three aspects. GO is used to annotate each of them. GO and the annotation of GO to genes (GO annotation, GOA) are two different things and are often confused. As can be seen from the fact that GO originally started with eukaryotes (and mainly animals), there are parts of GO that do not correspond to the actual situation with respect to other organisms, and various ontologies have been created for each domain (https://obofoundry.org (accessed on 1 April 2025)). Nonetheless, GO is very often used. It is because the core parts, such as energy metabolism, are common to all organisms. Therefore, GO is often annotated when genomes are newly sequenced in many organisms.

To carry out functional annotation, the first step is to predict the protein-coding sequences (abbreviated as coding sequence, and often called CDS) in the gene sequence and obtain their amino acid sequences [24,25,26]. The predicted protein sequences are then used to perform sequence homology searches on protein sequence sets of other species. Here we can introduce a functional annotation workflow (Fanflow), which is the method used initially for insects [34,35]. Fanflow first performs functional annotation using sequence similarity search and protein domain search for all protein sequences translated from cDNA sequences obtained by transcriptome analysis (Figure 4A). Annotation of non-coding RNA sequences obtained at the same time is also attempted using sequence similarity search and RNA domain search. Furthermore, the expression level information obtained from transcriptome analysis is integrated and used as functional annotation information. Using this workflow, have we performed functional annotation on sequences obtained from stick insect and silkworm transcriptomes and have obtained richer functional annotation information than previous studies on silkworms (Figure 4B) [35]. The suite of programs developed for this purpose are available as open source software on GitHub (https://github.com/bonohu/SAQE (accessed on 1 April 2025)).

### 3.4. Data Interpretation

The sequenced genome sequence information and the numerous annotations attached to it are meaningless unless they are utilized. Even in 2025, a quarter of a century after the complete genome information of living organisms became decipherable, it is still a matter of research just to find ways to utilize the information.

It is difficult to decipher the genome sequence of a single species by simply looking at it. This is because, unlike automobiles, living organisms are not structures that have been designed by humans in advance. Therefore, comparative genome analysis is used as an effective tool. In particular, differential analysis by comparing closely related species or different individuals within the same species is effective. Take, for example, the comparison of pathogenic *E. coli* O157 and non-pathogenic *E. coli*. Comparing the genome sequences of the two species, we can see that the difference is what is responsible for the pathogenicity. For example, the comparison between insecticide-resistant and insecticide-sensitive bed bugs revealed gene mutations likely conferring insecticide resistance. Such comparative genome analyses will increase in the future. However, even if bioinformatics can identify candidates, it is essential to verify these candidates in “wet” experiments. This is where genome editing comes in.

## 4. Genome Editing

The use of genome sequences and their annotation information obtained from genome sequencing is not limited to academic research. Industrial applications are possible, as described in the special issue *High-Throughput Sequencing Data Analysis for Industrial Applications* (https://www.mdpi.com/journal/ijms/special_issues/ID21EF7OZD (accessed on 1 April 2025)). One of the outstanding applications is genome editing. First, an overview of genome editing will be given, and then examples of genome-edited food products that have already been published in Japan will be introduced, showing which genes are targeted and how they are modified. Finally, we will discuss how to select target genes for genome editing and what kind of mutations to introduce into them in order to design them.

### 4.1. What Is Genome Editing

Genome editing is a technology that aims to alter the genomic DNA sequence of an organism [4]. Nucleases, which act as scissors, break the double strand of DNA, and the genome sequence itself is altered by taking advantage of the errors that occur when DNA is repaired by the organism’s original mechanism. Genome-editing technologies are categorized as shown in the table below, and while the introduction of foreign DNA is possible depending on the method used, all genome-editing technologies in practical use described below are based on gene knockout, which artificially induces mutations that occur in nature at a targeted location.

The difference between genome editing and genetic modification is that genome editing aims to cause a specific mutation at a specific position on the genome, while genetic modification aims to introduce a gene into the genome of another organism, thereby imparting new characteristics to that organism.

CRISPR–Cas9, where CRISPR stands for clustered regularly interspaced short palindromic repeats and Cas9 for CRISPR-associated protein 9, is widely used as a genome-editing tool. Its license is still under patent dispute in 2025, making it difficult to use it for commercial purposes. TALEN (transcription activator-like effector nuclease) and ZFN (zinc finger nuclease), which have been used before CRISPR–Cas9, are still in use because they are less likely to be off-targeted. ZFN is currently attracting attention because its patent has expired.

### 4.2. Genome-Edited Organisms

In Japan, genome-edited foods have already been published, as summarized in Table 2.

In addition to the organisms listed in this table, another example in animals that is expected to be put to practical use is in chickens that lay allergen-reduced eggs [36]. This is achieved by modifying and knocking out the ovomucoid gene that encodes the ovomucoid protein, which is one of the proteins responsible for egg allergies and which cannot be removed by heat treatment. This should be of great benefit to people with egg allergy, as it enables them to use vaccines and egg products that were previously unavailable to them.

### 4.3. Genome Editing Target Design

Various bioinformatic tools have also been developed [4]. Such tools published in the special issue *Research Advances in the Bioinformatics of Genome Editing and Gene Function Analysis* include designing targets to guide RNA effortlessly for target-AID purposes [37] and risk prediction of RNA which is off-target from base editors [38]. The genome-editing targets mentioned above have been used for genes that have been known from many years of research and are able to restore suppressed functions or disable the synthesis of allergenic proteins when the gene is knocked out. In the future, it will be necessary to find such genes through large-scale genome analysis and data accumulated so far in public DBs. We will introduce how to select new target genes for genome editing by using such data.

#### 4.3.1. Large-Scale Genome Analysis Results

Genome-wide association studies (GWASs), a method of genetic statistical analysis that examines the relationship between genetic variation (mainly single nucleotide polymorphisms) scattered throughout the human genome and trait information (characteristics of specific diseases or physical traits, such as susceptibility to cancer or susceptibility to alcohol) by collecting genome information from many people, is widely used in humans. The results are stored in a database called GWAS Catalog (https://www.ebi.ac.uk/gwas/ (accessed on 1 April 2025)), which can be freely accessed by anyone free of charge.

AlphaMissense is an AI tool developed by Google DeepMind to predict the harmfulness of genetic mutations [39]. Specifically, it analyzes missense mutations (mutations in which a DNA base sequence is altered or replaced, resulting in a change in the amino acid sequence and the creation of an abnormal protein) and predicts the likelihood that they will cause disease. It is estimated that 89% of the approximately 71 million missense mutations that can occur in the human genome can be classified. It was designed based on AlphaFold, a protein structure calculation AI, and incorporates a neural network called a “protein language model” trained on millions of protein sequences. Predictions in AlphaMissense are publicly available, which can provide scientists with information that can help identify the causes of inherited diseases. However, when using AI tools such as AlphaMissense, it is important to properly evaluate and carefully handle its results.

#### 4.3.2. Public Transcriptome Data

Compared to genome sequences, transcriptome data are not unique to each organism but vary with developmental stages and tissues. For example, a group of genes that fluctuates due to heat stress is not the same from one research group to another. The differences can be attributed to differences in experimental conditions and environments, as well as to differences in data analysis methods. While the former cannot be helped, the latter can be aligned by unifying data analysis methods (meta-analysis) [40].

Therefore, meta-analysis of the transcriptome of various stress stimuli from public databases has been attempted (Table 3). Enrichment analysis is used for genes whose expression is found to be altered in the meta-analysis. As a result, genes whose expression changes are observed in many experiments can be used as candidates for genome-editing targets as genes related to the stress stimuli.

#### 4.3.3. Bibliographic Data

Medical and biological articles are stored in PubMed as literature data. Bibliographic data of full-text articles are available at PubMed Central (PMC). At the time of writing at the end of February 2025, 38 million and 10 million articles are registered in PubMed and PMC, respectively (https://www.ncbi.nlm.nih.gov/search/all/?term=ALL%5Bfilter%5D (accessed on 1 April 2025)). The totality of bibliographic data is referred to as the ‘bibliome’, and bibliome analysis is a method of analyzing all of these data or bibliographic information. An example of this is the search for novel hypoxia-responsive genes by bibliome analysis [41]. The number of PubMed articles and the similarity (Simpson’s coefficient) of the HIF-1A gene of the HIF-1 transcription factor to the 100 genes whose expression is upregulated by hypoxic stimuli, which was obtained by the above meta-analysis of public transcriptome data, can be visualized as a scatterplot. Four genes (*GPR146*, *PPP1R3G*, *TMEM74B*, and *PRSS53*) were identified as novel hypoxia-responsive genes by visualizing them.

#### 4.3.4. Pathway Data

Pathway DBs such as KEGG [52] and Reactome [53] are often used as a curated set of genes for enrichment analysis, in addition to GO. They are used to perform enrichment analyses using the information of gene-encoding enzymes used in specific pathways (e.g., the glycolytic pathway and cholesterol synthesis pathway), using the variation in the expression of only those genes in aggregates. However, it is important to not only use the information of how a gene appears in a particular pathway, but also to use information about where it was located in the pathway. When a pathway of interest is found in the enrichment analysis, one should look at the pathway diagram to see where those genes are mapped in that pathway.

There are also many pathways that are not in the pathway DB mentioned above. If that pathway diagram is already open access and listed in the PMC, it would be registered in pathway figure OCR (PFOCR) [54]. PFOCR is an open science project that aims to extract pathway information from PMC and make it freely available to everyone. PFOCR utilizes artificial intelligence (AI) for extracting gene names and chemical compounds from figure images in PMC. An enrichment analysis tool using PFOCR pathway data has also been developed and made publicly available (https://github.com/gladstone-institutes/Interactive-Enrichment-Analysis (accessed on 1 April 2025)).

As a result, once the target pathway is known, a custom pathway diagram can be created by adding to the existing pathway diagram and placing genes in it; the expression profiles of the gene clusters belonging to it can then be viewed individually. For this purpose, the WikiPathways (https://www.wikipathways.org (accessed on 1 April 2025)) system, which allows pathway data to be shared like Wikipedia articles, can be used [55]. By publishing the new pathway diagrams created by the author, public web tools will be available and various data analyses can be performed using them. A system that facilitates the analysis of customized pathway diagrams using the WikiPathways platform, called Quest for Pathways with eXpression (QPX), has been developed to perform pathway analyses on non-model organisms (Figure 5) [56,57].

## 5. Conclusions and Future Perspectives

As a result of all of the research to date, the decoding of individual genomes has revealed mutation information on various phenotypic systems. There is concern that genome-editing technology may lead to the modification of human beings themselves. On the other hand, several general-purpose computational tools using generative AI are rapidly being developed. Harmonious tools that make the best use of these achievements will be greatly needed soon.

## Figures and Tables

**Figure 1 ijms-26-03442-f001:**
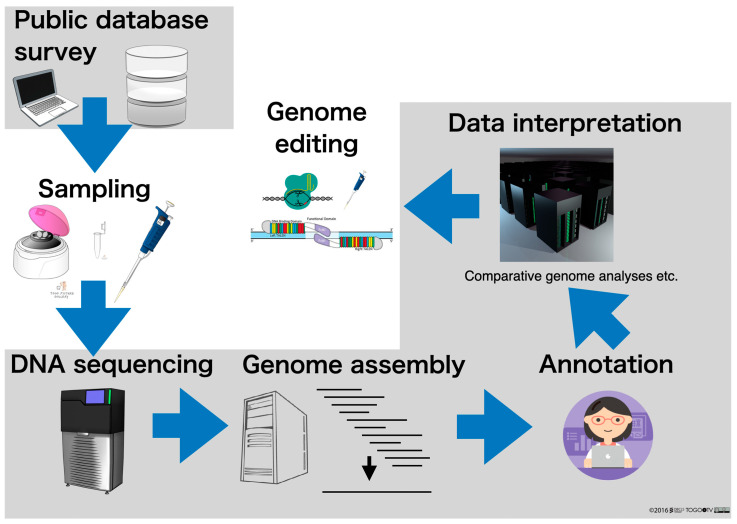
Strategies for genome editing, including genome sequencing. The gray background is the part with emphasis on bioinformatics.

**Figure 2 ijms-26-03442-f002:**
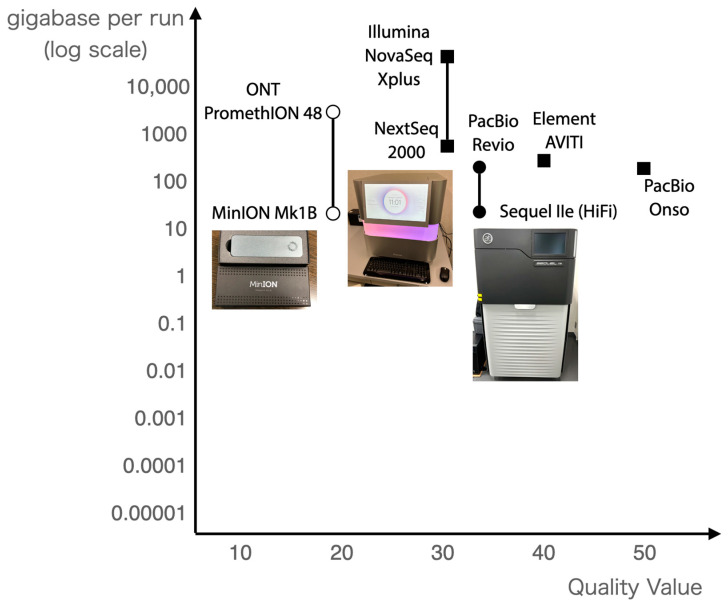
Next-generation sequencers in use in 2025. The horizontal axis shows the quality value, and the vertical axis shows the number of bases per run.

**Figure 3 ijms-26-03442-f003:**
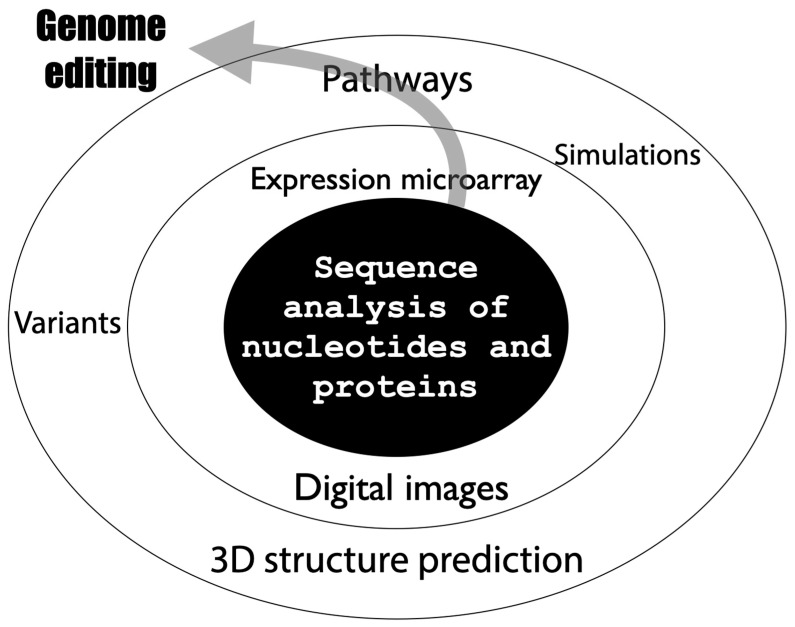
Expanding scope of bioinformatics.

**Figure 4 ijms-26-03442-f004:**
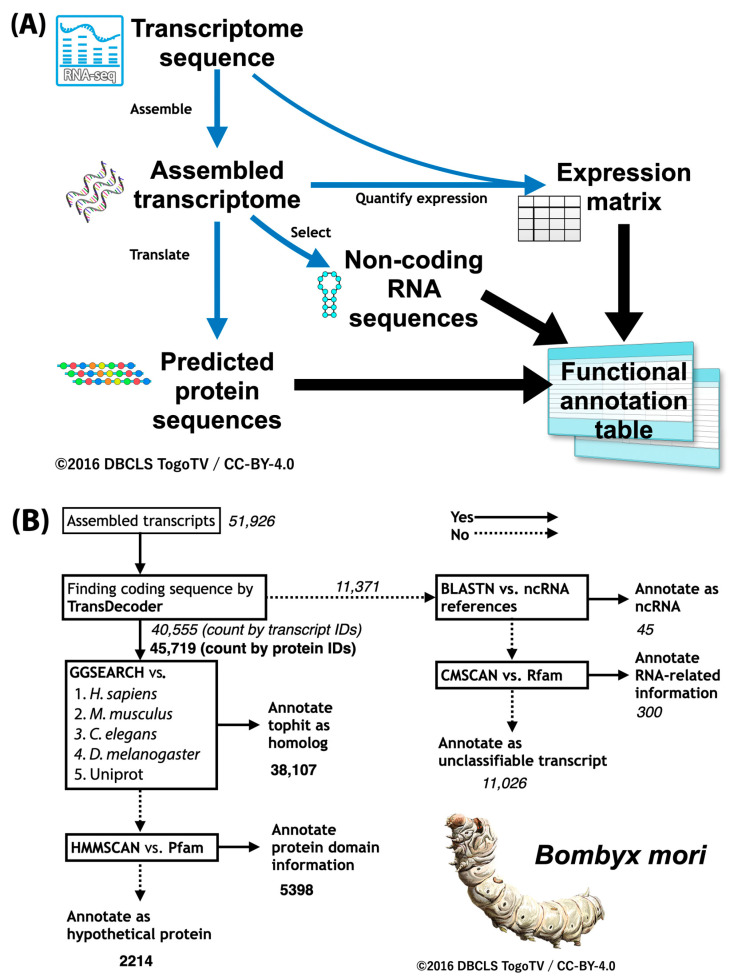
Fanflow: Functional annotation workflow. (**A**) Overall workflow in Fanflow. (**B**) Practical results for silkworm (*Bombyx mori*) by Fanflow. Fanflow uses sequence analysis and expression information to functionally annotate not only all protein sequences translated from cDNA sequences obtained by transcriptome analysis, but also all non-coding sequences obtained.

**Figure 5 ijms-26-03442-f005:**
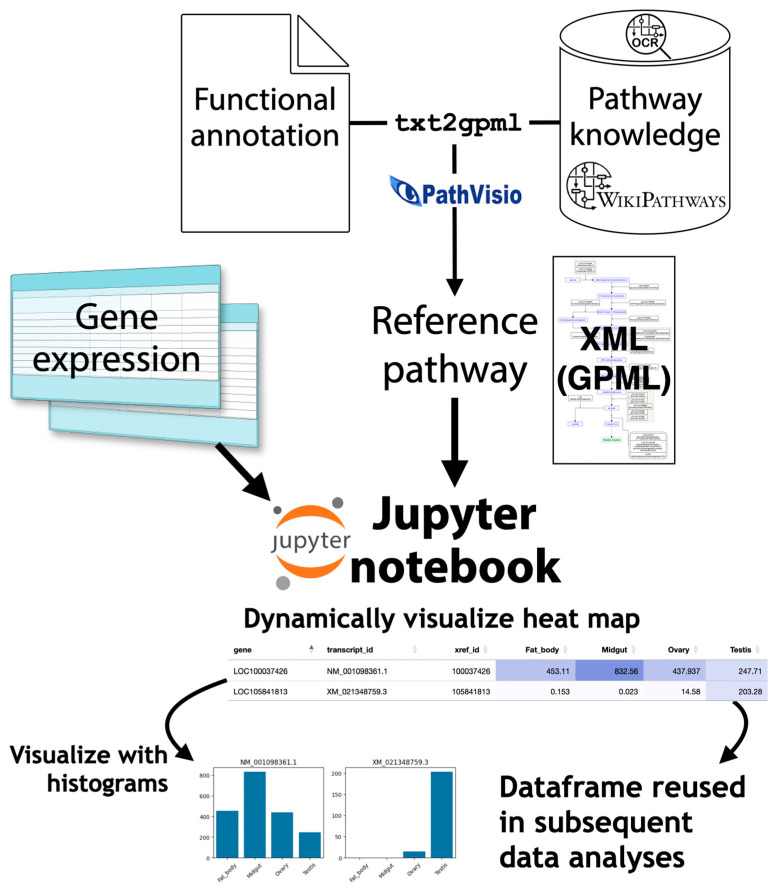
Quest for Pathways with eXpression (QPX). From the functional annotation of newly sequenced organism and pathway data from WikiPathways and PMC, QPX can generate customized pathway diagram data in Graphical Pathway Markup Language (GPML) and visualize gene expression data on pathways using the Jupyter notebook.

**Table 1 ijms-26-03442-t001:** Practical statistics on genome sequencing in the Hiroshima University Genome Editing Innovation Center.

Species Name	Scientific Name	Method	Genome Size (Mb)	Total Base (Gb)	X
Insecticide-resistant bed bug	*Cimex lectularius*	PacBio HiFi	615	20.1	32.7
Insecticide-sensitive bed bug		PacBio HiFi	645	22.3	36.2
Japanese parasitic wasp	*Copidosoma floridanum*	PacBio HiFi	553	28.1	50.8
Beefsteak plant (red perilla)	*Perilla frutescens*	PacBio HiFi + Hi-C	1259	72.4	57.5
Oriental armyworm	*Mythimna separata*	PacBio CCS + short reads	682	127	187
Edible green alga	*Ulva prolifera*	ONT + short reads	104	17.2	166

CCS: circular consensus sequencing, ONT: Oxford nanopore technologies.

**Table 2 ijms-26-03442-t002:** Agricultural, forestry, and fisheries products for which a letter of information was submitted to the Ministry of Agriculture, Forestry, and Fisheries. If the variety or lineage is different before being modified by genome-editing technology, it is listed as a separate organism. Information that may cause undue advantage or disadvantage to a specific person if made public is excluded. NA: not available. Originally from https://www.maff.go.jp/j/syouan/nouan/carta/tetuduki/nbt_tetuzuki.html (accessed on 1 April 2025) (Japanese, and the translation was carried out by the author).

Organism	Lineage	Information Provider	Date of Information Provided	Start Date of Use	Scheduled Date of Sales Launch
GABA-rich tomato	#87-17	Sanatech Life Science	2020-12-11	2020-12	2021-04
Increased edible portion red seabream	E189-E90	Regional Fish	2021-09-17	2021-09	2021-10
	E361-E90	Regional Fish	2022-12-06	2022-12	2023-01
High-growth tiger puffer	4D-4D	Regional Fish	2021-10-29	2021-10	2021-11
	Traditional lineage-4D	Regional Fish	2022-12-06	2022-12	2023-01
Waxy maize	PH1V69 CRISPR–Cas9	Corteva Agriscience Japan	2023-03-20	NA	NA
GABA-rich tomato	#206-4	Sanatech Life Science	2023-07-27	NA	NA
High-growth flounder	8D	Regional Fish	2023-12-25	2023-12	2024-04

**Table 3 ijms-26-03442-t003:** Meta-analysis of public transcriptome data conducted in the Hiroshima University Genome Editing Innovation Center.

Organisms	Stresses	Manuscript DOI
Human	Hypoxic	[41,42]
Human, Mouse	Oxidative	[43]
*D. melanogaster*, *C. elegans*	Oxidative	[44]
*A. thaliana*, *O. Sativa*	Hypoxia	[45]
Insects	Crowding	[46]
Insects	Caste	[47]
Human, Mouse	Heat	[48]
*A. thaliana*	Abiotic	[49]
Pig, Chicken	Breeding	[50]
Fishes	Gender	[51]

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
