# Peer review of "Recent Advances in Genome Editing and Bioinformatics: Addressing Challenges in Genome Editing Implementation and Genome Sequencing"

_ijms, 2025, doi:10.3390/ijms26073442_

Round 1
Reviewer 1 Report
Comments and Suggestions for Authors
I have the following comments;
- The manuscript is about Bioinformatic Approach to Exploring Genome Editing Targets 2
through Pathway Design but the author did not focus on the title of the manuscript and describe some of the other aspects. - The author should focus of the bioinformatics tools and describe how it can help to identify the targets for genome editing and how off-target effects can be minimized.
- Some of the sections need to be improved and expand and require more literature background and critical viewpoint.
- The figures needs to be improved as some of the figures are very simple and not conveying any significant knowledge about the topic.
- Also, there are some typos.
Author Response
Comments:
> I have the following comments;
> 1. The manuscript is about Bioinformatic Approach to Exploring Genome Editing Targets 2
through Pathway Design but the author did not focus on the title of the manuscript and describe some of the other aspects.
> 2. The author should focus of the bioinformatics tools and describe how it can help to identify the targets for genome editing and how off-target effects can be minimized.
Response:
Thanks for your helpful comment. As I wrote the review, my perspective changed from pathway design to the implementation of genome editing and genome sequencing that preceded that, I will change the title to “Recent Advances in Genome Editing and Bioinformatics: Addressing Challenges in Genome Editing Implementation and Genome Sequencing”. The references are revised to reduce the manuscript citations in Table 4 (currently Table 3).
Comments:
> 3. Some of the sections need to be improved and expand and require more literature background and critical viewpoint.
> 4. The figures needs to be improved as some of the figures are very simple and not conveying any significant knowledge about the topic.
Response:
Thanks for your comments. While the comments are so general that I cannot understand what part of the manuscript the reviewer is talking about, I reconsider the organization of figures and tables in the manuscript. According to reviewer’s comment, table 2 in the original version was removed to focus much on the main theme. I’m sure figures in the manuscript are original and informative enough for the reader of IJMS.
> 5. Also, there are some typos.
I recheck the manuscript and tried to fix the typos in the manuscript thoroughly.
Reviewer 2 Report
Comments and Suggestions for Authors
In this manuscript, the authors summarized genome sequencing and bioinformatics technologies. More about pathway design needed to be added.
- As mentioned in the title, this is a review related to bioinformatic approach and pathway design. However, only a short paragraph is about pathway design. Can add more about this part, such as metabolic model and others.
- AI technology have been applied in pathway design. Provide more samples.
Author Response
Comments 1:
> In this manuscript, the authors summarized genome sequencing and bioinformatics technologies. More about pathway design needed to be added.
> 1. As mentioned in the title, this is a review related to bioinformatic approach and pathway design. However, only a short paragraph is about pathway design. Can add more about this part, such as metabolic model and others.
Response 1:
Thanks for your helpful comment. As I wrote the review, my perspective changed from pathway design to the implementation of genome editing and genome sequencing that preceded that, I will change the title to “Recent Advances in Genome Editing and Bioinformatics: Addressing Challenges in Genome Editing Implementation and Genome Sequencing”.
Comments 2:
> 2. AI technology have been applied in pathway design. Provide more samples.
Response 2:
Thanks for your comment. I added one more sentence for the description of the use of AI in PFOCR in Line 574.
> PFOCR utilizes artificial intelligence (AI) for extracting gene names and chemical compounds from figure images in PMC.
Round 2
Reviewer 1 Report
Comments and Suggestions for Authors
The manuscript has been revised and improved significantly.